# Alcohol policy measures are an ignored catalyst for achievement of the sustainable development goals

**Kristina Sperkova** [1] *, **Peter Anderson** [2,3]ʘ, **Eva Jané Llopis** [2,4,5]ʘ

**1** Movendi International, Stockholm, Sweden, **2** Department of Health Promotion, CAPHRI Care and Public Health Research Institute, Maastricht University, Maastricht, The Netherlands, **3** Population Health Sciences Institute, Newcastle University, Newcastle upon Tyne, United Kingdom, **4** University Ramon Llull, ESADE, Barcelona, Spain, **5** Institute for Mental Health Policy Research, CAMH, Toronto, Canada

ʘ These authors contributed equally to this work.
* kristina.sperkova@gmail.com

**Data Availability Statement:** All relevant data are within the paper and its Supporting Information files.

**Funding:** The author(s) received no specific funding for this work.

## Abstract

### Background

By adopting Agenda 2030, governments agreed to review and report on their approach and action for achievement of sustainable development goals annually through the High-Level Political Forum (HLPF) on Sustainable Development. Health and well-being are at the heart of the United Nations Agenda 2030. Given the social and economic harm that can be done by alcohol, reducing the consumption of alcohol is a pre-requisite to achieve the sustainable development goals. We explored how selected European countries have considered alcohol-related harm as an obstacle to achievement of SDGs and the extent to which they view alcohol policy as a solution to the achievement of sustainable development by analysing their voluntary national reviews (VNRs) submitted to the HLPF between years 2016 and 2020.

### Methods

We developed our own framework with 260 questions reflecting three dimensions of alcohol-harm considerations: indication, action, and evaluation. We analysed 36 VNRs of 32 European countries by first assessing them against the 260 questions to find out how they report on alcohol harm and whether they, in their action, refer to evidence-based, cost-effective alcohol policy solutions. Afterwards we used content analysis to assess the extent to which the countries addressed alcohol related harm, whether they refer to alcohol harm within SDG 3 (good health and well-being) or look beyond the health goal and consider alcohol harm having impact on goals other than the Goal 3.

### Findings

Nine countries (28.1%) did not mention alcohol in their report. Only eight countries (25%) mentioned one or more of the alcohol policy best buys among the actions they are taking to reduce alcohol related harm and only three (9.3%) explicitly elaborated on their impact on

**Competing interests:** The authors have declared that no competing interests exist.

goals other than goal 3. Only five countries referred to the agreed indicator 3.5.2 measuring alcohol per capita consumption in the adult population. Many of the remaining countries used a range of terminology rather than alcohol per capita consumption, including "excessive use of alcohol", "heavy use", "too much alcohol ", "harmful alcohol consumption", "use among young people".

## Interpretation

Alcohol use is, for example, associated with violence (SDG 5 and 16), it contributes to inequalities (SDG 5 and 10), it hinders economic growth (SDG 8), disrupts sustainable consumption (SDG 12) and it adversely impacts environment (SDG 13 and 14). The findings of this study show that these effects are not considered in the design of measures to achieve these goals. Moreover, inaccurate language related to alcohol harm indicates a gap in understanding of extend of alcohol burden and the consequences for sustainable development. So does the choice of ineffective measures to reduce alcohol consumption. Education programs and awareness raising campaigns focusing on individual lifestyle are neither in line with WHO Global Strategy to reduce the harm caused by alcohol that all selected countries adopted in 2010, nor do they reflect the seriousness of the problems related to alcohol use. Effective alcohol policy measures, so called three best buys, are missing from the transformative action that the Agenda 2030 calls for and governments committed to.

## Introduction

Alcohol is a major risk factor for ill-health and premature death. In 2016, it was responsible for 5.1% of the global burden of disease, and three million deaths (5.3% of all deaths) [1]. The European region has the highest total alcohol per capita consumption and the highest share of all deaths attributable to alcohol compared with other risk factors. Alcohol accounts for 7.1% and 2.2% of the global burden of disease for men and women respectively and is the leading risk factor for premature mortality and disability among those aged 15 to 49 years, accounting for 10% of all deaths in this age group [2]. People of low socioeconomic status are disproportionately affected [3]. In 2010, the World Health Assembly endorsed the Global Alcohol Strategy [4] to guide governments in the formulation and implementation of effective national alcohol policies, including the three most effective interventions, the so-called best buys: increasing alcohol excise taxes, bans or comprehensive restrictions on alcohol advertising, and restrictions on alcohol availability. Since the adoption of the Global Strategy, alcohol consumption decreased in 34 countries, but increased in 17 countries of the WHO European region [5].

Health and well-being are at the heart of the United Nations Agenda 2030 [6]. The harm due to alcohol is an impediment to reaching at least 14 out of 17 the sustainable development goals (SDGs) [7]. Given the social and economic harm caused by alcohol [8], reducing alcohol consumption and related harm is a catalyst for the SDGs, highlighted by target 3.5 *Strengthen the prevention and treatment of substance abuse, including narcotic drug abuse and harmful use of alcohol* [9]. Effective and comprehensive alcohol policy is necessary for achieving SDG 3 *Ensure healthy lives and promote well-being for all at all ages* but also other goals. The available information about alcohol harm and sustainable development goals presents three main categories of scientific analysis of the intersection of the two areas: a) a description of the problem by exposing the intersection of alcohol harm and achievement of sustainable development [7];

b) an emphasis of the importance of policy coherence, including alcohol policy, for achievement of sustainable development goals [10]; and c) an analyses of the reasons for inconsistency in policy making [11] such as conflict of interest and alcohol industry interference. Although there are several papers that write about alcohol and sustainable development, the focus lays solely on the goal three, while a systematic overview of how governments currently utilize the potential of alcohol policy solutions for achievement of sustainable development goals does not yet exist.

The SDGs are a part of the 2030 Agenda for Sustainable Development adopted by the General Assembly of the United Nations in September 2015. In the Agenda 2030 governments state their determination to take bold steps for transformative change to shift the world on to a sustainable and resilient path [6]. The 17 SDGs and 169 targets form a concrete plan to end poverty, protect all aspects of the planet's habitability and ensure that all people enjoy peace and prosperity [12]. The 17 goals are universal, integrated, and indivisible [6]. By adopting the Agenda 2030, governments promised to shift their approach and address the economic, social, and environmental dimensions of sustainable development in cross-sectoral and cross-cutting manners.

The SDGs are legally non-binding, and their effective implementation depends largely on the good will of national governments [13]. Governments agreed to review and report on their approach and action for the achievement of the SDGs annually through the High-Level Political Forum on Sustainable Development (HLPF), which was mandated in 2012 through the United Nations Conference on Sustainable Development (Rio+20) [14]. One pillar of the Forum is an in-depth review of selected SDGs. The reviewed SDGs differ from year to year, and all 17 SDGs are covered within a three-year cycle set by a Resolution of the General Assembly [14]. Another pillar follows up country action through voluntary national reviews (VNRs). Each country is expected to report at least once in a four-year cycle. Two hundred forty-seven VNRs were submitted to the Forum between 2016 and 2021 [14].

There is no template for VNRs, although common reporting guidelines are issued by the Secretary General [15] recommending concrete chapters. Most countries examine actual progress on all targets (and indicators). In an analysis of 20 European VNRs, it was concluded that whilst robust numerical data on progress might be presented, information on the actions that countries have taken to achieve the goals is often lacking [16]. Another approach to reporting is to document potential interactions between different SDGs, and to consider the positive and negative effects of these, including the ways to enhance and reduce the effects respectively.

Since failure to address alcohol-related harm impedes progress towards the SDGs, the aim of this paper is to explore how selected European countries have considered alcohol-related harm as an obstacle to achieving the SDGs and the extent to which they view alcohol policy as a solution for promoting sustainable development. Our hypothesis is: In their reporting process to the HLPF (VNRs), governments do not consider the impact of alcohol policy solutions on the achievement of the SDGs.

We ask the following questions:

1. Do governments mention alcohol harm in their VNRs to the HLPF?

2. Do governments mention alcohol policy solutions in their VNRs to the HLPF?

3. Do governments address the impact of alcohol policy on the achievement of the SDGs in their VNRs to the HLPF?

## Methodology

To answer the research questions, we searched for the word '*alcohol*' in all available VNRs from all selected countries and coded the answers. We analysed all 36 VNRs [17] submitted to the HLPF [14] by 32 countries– 27 EU, three EEA countries, the UK and Switzerland–between 2016 and 2020. To ensure the replicability of the analysis, we developed a new framework. The framework was created after we had initially screened all VNRs to identify how the countries reported on alcohol consumption, alcohol harm and alcohol policy. Based on the basic understanding of the countries' approach to address alcohol in their reviews, we created a coding system for the development of the framework dimensions (see S1 File).

The final framework consisted of 260 questions divided into three dimensions that reflect the level of recognition of alcohol-related harm in each country's VNR. The dimensions are Indication, Action and Evaluation. After the word search for *alcohol* in all available VNRs from all selected countries, the search results in the form of text excerpts containing the word *alcohol* were inserted into the framework.

Besides the three dimensions, the framework included information about the level of alcohol harm in each country based on years of life lost (YLL) ranked by WHO [18] and the specific structure a country chose to use to report on their implementation of Agenda 2030 (reporting on all goals one by one, or their own reporting scheme) [15].

### Quantitative analysis

All VNRs were assessed against the 260 questions and the three dimensions: *Indication*–no or minimal mention of solely quantitative data about alcohol harm or alcohol consumption in the country; *Action*–reference to measures the countries take to address alcohol related harm; and *Evaluation*–exploration of the impact of the actions taken on the achievement of the SDGs. Each reference to alcohol regardless of the question or dimension received a score of one. When a review did not provide an answer to a question out of the set of 260 questions, it scored zero on that particular question. A country review that received at least one score in the whole framework is counted as a country that, to some extent, recognizes the need to address alcohol in the implementation of Agenda 2030.

### Qualitative analysis

**Level of recognition.**  After selecting the countries that scored at least once in the whole framework, we analysed the level of their recognition of alcohol-related harm being an impediment to the achievement of the SDGs. We used content analysis to assess the extent to which the countries addressed alcohol harm in the VNRs. We examined whether they stated any solutions to the harm caused by alcohol (Action dimension) and whether they referred to the impact that alcohol harm or the introduced solutions have on the achievement of the SDGs (Evaluation dimension).

In the *Action* dimension, we examined the level of effectiveness of the tools and actions used to reduce alcohol harm in the VNRs. We considered whether the countries reported on the implementation of the alcohol policy best buys or suggested other, less effective, interventions such as awareness campaigns.

In the *Evaluation* dimension, we focused on ways countries elaborate on the impact of alcohol harm and/or the introduced actions (policies, campaigns, strategies) on the achievement of the SDGs.

**Location of the problem and solutions.**  In the content analysis, we also distinguished between alcohol-related harm, actions and solutions being placed under SDG 3 (where they naturally belong due to target 3.5. and indicator 3.5.2) or being mentioned under other goals.

Locating reflections about alcohol harm under goals other than SDG 3 would suggest that alcohol is being treated as a cross-cutting issue. In this step we also considered the structure each country has chosen for their VNRs.

# Results

## Quantitative analysis

**Countries and their VNRs.**   We included all 27 countries of the European Union, three countries from the European Economic Area, Switzerland and the UK. We accessed 37 VNRs of 32 countries submitted to the HLPF by December 2020. All VNRs were submitted in English except that of Luxemburg, which was submitted in French. Four countries have reported to the HLPF two times within the period 2016 and 2020. All the other countries have so far reported only once, Table 1.

The length and structure of the VNR reports varied. The shortest report of 28 pages was written by Switzerland, followed by Norway with 29 pages. The longest report of 300 pages was submitted by Ireland. The second longest VNR of 235 pages was presented by the UK. The median number of pages was 88.

Nine countries (28.1%) did not mention alcohol in their reports, Table 1.

**Indication dimension.**   Fourteen countries (43.7%) reported statistical information about alcohol use or alcohol related harm, Table 2. One of these 14 countries (Austria) did not elaborate further either on their actions to reduce alcohol consumption or the impact reduction of alcohol use would have.

**Action dimension.**   Twenty-two countries (68.6%) reported on action to reduce alcohol use or/and alcohol-related harm, of which nine (28%) specifically named one or more of the alcohol policy best buys, Table 2.

**Evaluation dimension.**   Eleven countries (34.3%) elaborated on the impact of alcohol-related harm or the actions they have introduced and reported in VNRs, of which five outlined the impact of the alcohol policy best buys on achieving the SDGs, Table 2.

**Beyond SDG 3 –General.**   Eleven countries (34.3%) connected alcohol-related harm, actions to address the harm and/or the evaluation of their impact to goals other than SDG 3, Table 2.

**Beyond SDG 3 –Action.**   Altogether, nine countries (26.4%) mentioned action on alcohol under goals other than SDG 3, including goals 2, 4, 8, 9, 10 and 12. Of these countries, Finland and Hungary mentioned the importance of an alcohol policy best buy solution and all other countries wrote about actions from the category "other than the three best buys". The most frequently mentioned goal under the action category was SDG 10, mentioned by five out of nine countries, followed by goals 4 and 8, mentioned by two out of nine countries each, Table 2.

**Beyond SDG 3 –Evaluation.**   Eight countries (25%) countries evaluated the impact of their actions to reduce alcohol use on goals other than SDG 3. Estonia, Finland, Hungary explicitly and the UK implicitly analysed the impact of one or more of the alcohol policy best buys they have introduced on goals other than SDG 3 while the other countries have mentioned other actions. The most frequently mentioned goal the countries identified as adversely affected by alcohol use when they evaluated the impact of policies was SDG 10, mentioned five times, Table 2.

## Qualitative analysis

**Reporting differences on alcohol: APC, or "excessive" alcohol use..**   Countries differed in the way that they reported on alcohol in their VNRs. All countries primarily mentioned alcohol under SDG 3. Although the alcohol related indicator for target 3.5 used to be *3.5.2*

**Table 1.**

| Country | EU/EEA | Year of reporting (and SDGs in focus) | | | | |
|---|---|---|---|---|---|---|
| | | 2016 | 2017 (1,2,3,5,9, 14,17) | 2018 (6,7,11, 12,15, 17) | 2019 (4,8,10, 13,16,17) | 2020 |
| **Austria** | EU | | | | | x |
| **Belgium** | EU | | x | | | |
| **Bulgaria** | EU | | | | | x |
| **Croatia** | EU | | | | x | |
| Cyprus | EU | | x | | | |
| **Czechia** | EU | | x | | | |
| Denmark | EU | | x | | | |
| **Estonia** | EU | x | | | | x |
| **Finland** | EU | x | | | | x |
| France | EU | x | | | | |
| Greece | EU | | | x | | |
| Germany | EU | x | | | | |
| **Hungary** | EU | | | x | | |
| **Iceland** | EEA | | | | x | |
| **Ireland** | EU | | | x | | |
| **Italy** | EU | | x | | | |
| **Latvia** | EU | | | x | | |
| **Lichtenstein** | EEA | | | | x | |
| **Lithuania** | EU | | | x | | |
| Luxemburg | EU | | x | | | |
| **Malta** | EU | | | x | | |
| **Netherlands** | EU | | x | | | |
| Norway | EEA | x | | | | |
| **Poland** | EU | | | x | | |
| **Portugal** | EU | | x | | | |
| **Romania** | EU | | | x | | |
| Slovakia | EU | | | x | | |
| **Slovenia** | EU | | x | | | x |
| **Spain** | EU | | | x | | |
| **Sweden** | EU | | x | | | |
| Switzerland | none | x | | x | | |
| **UK** | none | | | | x | |

Overview of countries and their membership in EU/EEA and the years of reporting to HLPF; **Bold**–alcohol indicated in the report (regardless of the level of recognition).

*Harmful use of alcohol, defined according to the national context as **alcohol per capita consumption** (aged 15 years and older) within a calendar year in litres of pure alcohol* [9] *and after the comprehensive review by the UN Statistical Commission in the beginning of 2020 refined as Alcohol per capita consumption (aged 15 years and older) within a calendar year in litres of pure alcohol* [19] in the narrative parts of the analysed VNRs, countries used a range of terminology rather than alcohol per capita consumption, including "excessive use of alcohol", "heavy use", "too much alcohol", "harmful alcohol consumption", "use among young people", Table 3. Alcohol per capita is an indicator that all countries are expected to document. The trends in alcohol per capita consumption help to predict, interpret, and prevent harms caused

**Table 2.**

| Country | Indication | | | | | Action | | | | Impact evaluation | | | |
|---|---|---|---|---|---|---|---|---|---|---|---|---|---|
| | *Content* | | | Goal | | *Policy* | | Goal | | *Policy* | | Goal | |
| | Gen. | Harm | Cons. | 3 | Other | 3BB | Other | 3 | Other | 3BB | Other | 3 | Other |
| Austria (2) | - | x | x | x | - | - | - | - | - | - | - | - | - |
| Belgium (3) I | - | - | x | x | - | x | x | x | - | - | - | - | - |
| Bulgaria (4) I | x | - | - | x | - | - | x | x | - | - | - | - | - |
| Croatia (3) | - | - | - | x | - | - | x | x | - | - | - | - | - |
| *Czech Rep*** (3) I | x | - | - | - | - | - | x | x | - | - | - | - | - |
| Estonia (5) | - | - | x | x | - | x | x | x | - | x | - | x | **(8)**,10,11 |
| Finland (3) | x | - | x | x | - | x | - | x | **10** | x | x | x | **10** |
| Hungary (4) | - | - | - | - | - | x | x | x | **2** | x | - | x | **2** |
| Iceland* (2) I | x | - | x | x | - | x | x | x | - | - | - | - | - |
| Ireland (2) I | - | - | - | - | - | - | x | x | 8 | - | x | x | 8 |
| *Italy*** (1) | - | - | - | - | - | - | x | x | 4,8,10 | - | - | - | - |
| Latvia (5) | x | x | x | x | 17 | x | - | x | (10) | - | x | x | 4 |
| Lichtenstein* | x | - | - | x | - | - | x | x | - | - | x | x | - |
| Lithuania (5) I | x | - | x | x | - | x | x | x | (12) | x | x | x | - |
| Malta (1) I | - | - | - | - | - | - | x | x | 9 | - | - | - | - |
| Netherlands (2) | - | - | - | - | - | - | x | x | (10) | - | x | x | (10) |
| Poland (4) | - | - | - | - | - | - | x | x | - | - | - | - | - |
| Portugal (3) | - | - | - | - | - | (x) | x | x | 4,10,16 | - | x | x | 4,10,16 |
| Romania (4) | - | - | x | x | - | - | x | x | - | - | - | - | - |
| Slovenia (3) I | x | x | - | x | - | - | x | x | - | - | - | - | - |
| Spain (2) | - | - | | - | - | - | x | x | - | - | x | x | - |
| Sweden (2) | - | - | x | x | - | - | x | x | - | - | - | - | - |
| UK** (2) | x | - | x | x | - | x | x | x | - | x | x | x | **(10)** |

Overview of countries, that mentioned alcohol in their VNRs

* EEA countries

** non-EU/EEA countries

***countries with a different VNR structure; **Bold**–an SDG other than SDG 3 connected to a three best buy measure; I–has seen increase in alcohol consumption since 2010.

by alcohol while terms as "excessive" or "harmful" are not precisely defined and therefore difficult to compare and unhelpful for effective policy design.

Countries that mentioned alcohol per capita consumption were Austria, Estonia, Lithuania, Romania, and Sweden.

**Reporting differences on alcohol–alcohol consumption or alcohol-related harm.** Some countries did not report any levels of alcohol consumption but stated alcohol-related harm, such as premature mortality, suicides, and non-communicable diseases, Table 3.

**Action–Three best buys and other solutions.** Four countries (Belgium, Estonia, Latvia, Lithuania) reported a ban on or stricter regulations of alcohol advertising; four countries (Belgium, Latvia, Portugal and UK) reported on the regulation of the price of alcohol, such as taxation or minimum unit price; and, seven countries (Belgium, Estonia, Finland, Hungary, Iceland, Latvia, Lithuania) reported on their alcohol availability measures, such as intensifying control of vending machines, restrictions on display of alcohol beverages, or shorter hours of trade. In all cases, the three best buys were mentioned in sections dedicated to SDG 3, except for Hungary that referred to restricted alcohol availability to contribute to improved nutrition

**Table 3.**

| Differences in reporting on alcohol | |
|---|---|
| APC or excessive alcohol use | *". . .9% of the adult population consumes **too much alcohol**" (Belgium)* |
| | *"**Per capita use of alcohol** (litres of absolute alcohol per year per person) (Green Book on Alcohol Policy) 8.7 liters in 2015" (Estonia)* |
| Alcohol consumption or alcohol-related harm | *"Numerous **health problems** are directly related to (lack of) exercise, poor nutrition and obesity, as well as the consumption of nicotine and alcohol" (Lichtenstein)* |
| | *". . .alcohol**, consumption per capita** in 2014 and 2015 was at a relatively stable level of just over 9 litres of pure alcohol per inhabitant aged 15 and older" (Sweden)* |
| **Action** | |
| Three best buys and other solutions | *"**The Public Health Product Tax** also specifies that products listed in the Act on Public Health Product Tax (e.g., pre-packed sugary products, soft drinks, energy drinks, savoury snacks) and alcoholic and tobacco products shall not be sold at events organized for children and students." (Hungary)* |
| | *"**Awareness raising campaigns** in the fields of nutrition, exercise, nicotine and alcohol have shown that simple measures can achieve a great deal in terms of public health" (Lichtenstein).* |
| **Evaluation** | |
| Planned impact or real impact | *"**The purpose of alcohol policy** is to reduce the social, economic and health damages resulting from the abuse of alcohol. Also, to guarantee a supportive environment for the growth of children and youth and make the living environment safer for all." (Estonia)* |
| | *"**The Alcohol Act of 2017 increased the availability** of alcohol. Alcohol-related causes explain one fifth of differences in mortality by social group among Finnish men and one sixth among Finnish women. The new Alcohol Act (2017) increased the availability of alcohol, and the negative repercussions are felt especially in the low-income bracket, among the poorly qualified and long-term unemployed." (Finland)* |

Examples of concrete statements representing each category identified in the analysis of the VNRs.

among children and youth (SDG 2), and Finland that referred to alcohol availability and its impact on SDG 10, Table 3.

From among the other interventions presented by the countries, "awareness raising" occurred frequently, especially in connection to alcohol and pregnancy, to promotion of healthy lifestyle in general, and to reduction of chronic diseases in particular, Table 3.

**Evaluation–planned and real impact.** Reporting on impacts of the implemented actions took place on two levels: planned and real effects, Table 3. Reporting on planned impact means that a country mentioned scientific evidence of effects of alcohol policy measures but did not refer to any concrete results found in the country. It described rather the motivation to implement a certain measure (sometimes based on previous experience) than an evaluated effect, Table 3.

When countries reported on the level of the real impact, they shared effects of the implemented actions on SDG goals and targets.

**Extent of the impact: Impact on alcohol-related harm, impact on alcohol consumption, impact on well-being.** Regardless of the level of impact countries reported on, their VNRs differed in the specificity of effects reported, and on the extent of the impact.

Most countries reported the impact of alcohol policies or other actions they implemented within Goal 3. Ireland and Lithuania reported impact in the form of **reduction of alcohol use,** Table 4.

**Table 4.**

| Three best buys | | |
| --- | --- | --- |
| | **Anticipated impact** | **Real impact** |
| **Intermediate goal** | | Hungary (maybe alcohol too) |
| **Alcohol use** | | Finland, Latvia, Lithuania |
| **SDG 3** | | |
| **Other goals** | Estonia, UK | Finland, Latvia (4), Hungary |
| Other actions | | |
| | **Anticipated impact** | **Real impact** |
| **Intermediate goal** | | |
| **Alcohol use** | | Ireland |
| **SDG 3** | Netherlands, Spain | *Lichtenstein* |
| **Other goals** | Netherlands, Portugal | |

Overview of alcohol policy impact mentions in countries' VNRs, italics–impact is mentioned, but unclear what it is.

There was one mention of **intermediate impact** (goal) of introduced policies that has the potential to lead to reduced alcohol use and related harm (Hungary): reduced supply of products might lead to reduced alcohol use and better health, Table 5.

Some countries reported on **reduction of alcohol related harm** such as premature mortality. However, it is important to note that some VNR reports do not mention real impact but an anticipated or intended impact, Table 4.

Three countries reported "extended" **impact on other goals** through achieving SDG 3, meaning that the countries claim that positive developments in the health goal (better health and well-being) will influence other goals. The most frequently mentioned motivation (expected impact) for various strategies and actions is health promotion and well-being (SDG 3) and reduction of health inequalities (SDG 10). None of the actions mentioned in this category belonged to the three best buys (population-wide measures), Table 5.

Countries that have mentioned the impact of one of the three best buys on the achievement of goals other than SDG 3 were Estonia, Finland and possibly (but not explicitly) Hungary.

Finland was the only country that reported the **negative impact** of weak(ened) alcohol policy measures (increased availability), Table 5.

**Specific vs vague impact.** Out of the 12 countries that reported on impact of their policies, some countries, such as Finland, and the UK, were specific in the impact description, Table 5. Others were vague. Even though the impact is mentioned, it is not specified. Therefore, the concrete results of the actions remain unknown, only the value ascribed to them such as "great impact", "negative repercussion" point to intended effects, Table 5.

**Character of the report.** VNR reports that looked beyond SDG 3 and reported the impact of alcohol policy solutions or other actions reducing alcohol use and harm followed a "classic" reporting structure–goal by goal.

Italy, that has reported on *Action level* and looked beyond SDG 3, chose a different VNR structure in which they listed targets related to national priorities. Thus, they mentioned work on target 3.5 as a means (together with targets 1.3, 4.2, 5.1, 5.4, 16.6) to

*"Ensure the effectiveness of the social protection and security system"*

or (together with targets 10.2)

*"To fight deviance through prevention and social integration of vulnerable individuals"*

**Table 5.**

| Extend of the impact | |
|---|---|
| *Reduction of alcohol use* | *"Ireland was among several countries in the OECD to experience large **reductions in alcohol consumption** between 2000 and 2015, dropping from 14.2 litres per capita aged 15+ to 10.9 litres" (Ireland)* |
| *Intermediate impact* | *"The Public Health Product Tax, based on an impact assessment and in effect since September 2011, has reached its public health goals: **the supply and the turnover** of products with ingredients that have proven adverse effects on health **have decreased**". (Hungary)* |
| *Reduction of alcohol related harm* | *"The factors targeted in this Strategy are those considered most important from the standpoint of approaches to chronicity, such as healthy eating, physical activity, the consumption of tobacco and alcohol, emotional well-being and the safety of the environment (e.g., accident prevention). This approach represents an advance in comprehensive health intervention, strengthening primary disease prevention and health promotion, reinforcing interventions in universal primary care and promoting and coordinating community interventions in different environments (health, social, educational and community), with a view to **reducing premature mortality** from non-communicable diseases to minimal levels". (Spain)* |
| *Impact on other goals–"extended impact"* | *"The National Prevention Programme will continue to promote good health with a focus on health protection and reducing **health inequalities**". (Netherlands)* |
| *Impact on other goals mentioning three best buys* | *"The purpose of amending the Advertising Act is to protect public health, **reduce the social, economic, and health damage** caused by alcohol consumption, and ensure a **supportive environment for the growth and development for children and young people**". (Estonia)* |
| *Negative impact of weakening an alcohol policy measure* | *"The new Alcohol Act (2017) **increased the availability** of alcohol, and the **negative repercussions** are felt especially in the low-income bracket, among the poorly qualified and long-term unemployed". (Finland)* |
| Specific vs vague impact | |
| *Specific* | *"Building on previous work and policy initiatives such as minimum unit pricing for alcohol, the Alcohol Framework 2018: Preventing Harm has a key focus on **reducing health inequalities**". (UK)* |
| *Vague* | *"Awareness raising campaigns in the fields of nutrition, exercise, nicotine and alcohol have shown that simple measures can achieve a **great deal in terms of public health**". (Lichtenstein)* |

Examples of concrete statements representing each category identified in the analysis of the VNRs.

This method shows the interrelation between various targets and a need for a comprehensive approach to the main national priorities (not SDGs). However, it is difficult to understand from the report, which measures have been taken and what impact they actually had.

The SDGs in focus of a given year do not seem to strongly correlate with the extend of alcohol policy solutions consideration. The SDG 3 was in focus in 2017, but the countries that reported that year, do not seem to stand out in the level of elaboration on alcohol policy measures compared to reports from other years.

The level of elaboration on utilisation of alcohol policies for achievement of SDGs does not commensurate with the length of the report.

**No mention of alcohol.** Countries that did not mention alcohol at all in their VNRs used, except for Denmark, a classic reporting structure. Several of these countries elaborated in detail on various targets and even indicators but skipped indicator 3.5.2 and did not mention any information on alcohol.

Two countries (Norway and Switzerland) indicated their intention to focus their work on target 3.5, but did not mention any consumption statistics, harm nor alcohol related actions in their report.

## Discussion

### Main findings

**Alcohol policy best buys.** Considering the enormous potential that implementing the alcohol policy best buys and reducing overall alcohol consumption can have on achieving the SDGs, the number of countries that considered the effects of the alcohol policy best buys in their VNRs is very low.

Only eight countries (25%) mentioned one or more of the alcohol policy best buys among the actions they are taking to reduce alcohol related harm. Only five countries (15.6%) wrote about the impact of one or more of the three alcohol policy best buys on the achievement of the SDGs and only three (9.3%) explicitly elaborated on their impact on goals other than SDG 3. Hungary referred to restricted alcohol availability in order to contribute to improved nutrition among children and youth (SDG 2); Finland referred to alcohol availability and its impact on social inclusion and poverty (SDG 10); Estonia highlighted the importance of the alcohol policy best buys for the creation of safer environments (SDG 11) and mentioned the importance of effective alcohol policy for the reduction of economic harm caused by alcohol that could be interpreted as enabling of economic growth (SDG8). The UK extended the effect of better health and wellbeing on health equity that could be considered as an impact on the elimination of inequality (SDG 10).

These findings show that very few countries reflect on the adverse effects of alcohol related harm on the society and sustainable development that they have committed to in 2015. The findings also show that very few countries use population approach to alcohol related harm and most of the countries expect their citizens to take personal responsibility for their alcohol consumption. Overlooking population-wide solutions can mean a missed opportunity to implement highly cost-effective (and in the case of alcohol taxation even domestic resources mobilising) solutions and act for a transformative change to shift the world on to a resilient and sustainable path the countries had determined to do.

The vague language about the impact of alcohol policy measures to reduce alcohol harm suggests a gap in understanding of harms caused by alcohol and/or an absence of cross-sectoral analysis that would reflect the nature of alcohol harm adversely affecting SDGs beyond goal 3.

**Consideration of alcohol harm.** Although health is at the centre of Agenda 2030 and despite the strong relationship between health, wellbeing, and inequalities, consideration of this connection was rare. The choice of words indicates that governments continue to underestimate the full extent of alcohol harm and limit it to "alcoholism" and "excessive drinking". Limiting alcohol harm to "alcoholism" can be a partial explanation for the missing attention to the most cost-effective population level alcohol policy measures. Instead of utilizing the overall reduction of alcohol use as a catalyst for development, countries reported rather vague approaches to alcohol harm such as ineffective lifestyle campaigns or placing responsibility for alcohol harm solely on the individual. This framing prevents governments from seeing the broad impact of alcohol harm on education, equality, economic growth, and the environment. Governments are thus missing an opportunity to utilise the potential of alcohol policy best buys to accelerate progress towards multiple SDGs in a cost-effective and synergistic way.

The lack of consideration of the impact of alcohol consumption on the implementation of Agenda 2030 within the system of VNR reporting can have several reasons: a) weak understanding of alcohol-related harm and its impact on society, years of life lost, productivity, economic growth or overall safety; b) insufficient inclusion of all relevant sectors and stakeholders in the process of writing the VNRs; c) interference of alcohol industry in the process of writing

the VNRs; or, d) the nature of the reviews touching upon all SDGs without working through their interconnectedness.

In addition, there are a few countries generally known for evidence-based alcohol policy implementation, that have not reported on alcohol at all. That leads us to a reason e) a longer alcohol policy tradition that might lead to its invisibility thus the country's low engagement with alcohol policy as a catalyst for sustainable development.

To address the reasons, countries would need to a) better and systematically monitor the impact of alcohol production, consumption and related harm on the achievement of SDGs b) make sure that academics, civil society actors from the field of prevention and reduction of alcohol harm and public health agencies are actively involved in the national process of writing VNRs c) due to conflict of interest prevent alcohol industry involvement in formulation of VNRs in case of those SDGs that are adversely affected by alcohol related harm d) develop a reporting framework that enables to see how various goals relate to each and weight consequences of certain action or inaction on particular goals.

Regardless of the reasons, omitting the alcohol policy best buys and their impact on the achievement of the SDGs from VNRs means a missed opportunity not only to share experience, insights, and knowledge about how to use them as a catalyst for sustainable development but also to start building a collective understanding that these solutions could be a part of a basic action package for sustainable development. In Table 6 we provide examples of the adverse effects of alcohol on achievement of sustainable development goals:

Reduced consumption of alcohol leads to reduced alcohol related harm in all three dimensions of sustainable development–environmental, social and economic. Given current trends, UN targets for reduced alcohol consumption are unlikely to be met [20]. WHO established the potential of alcohol policy best buys for achieving the SDGs in their SAFER initiative [21]. The five SAFER strategies include the three alcohol policy best buys a) increasing excise taxes on alcoholic beverages, b) comprehensive restrictions on alcohol advertising, and c) restrictions on physical access to alcohol.

The implementation strategy of increased excise taxes includes an establishment of effective and efficient taxation system, adequate tax collection and tax enforcement. Increasing the price of alcohol is the single most effective strategy to reduce and prevent alcohol related harm. A 20% increase in the price of alcohol through higher taxes could accumulate as much as $9tn in increased revenues globally over 50 years [22] and could be a domestic source of financing for development (social, economic and environmental development).

Comprehensive bans or set of restrictions on alcohol advertising, sponsorship and promotion are the preferred measures in alcohol marketing regulation. Restricting only one aspect of the marketing mix, such as only promotion, or only product placement, often results in an expansion of alcohol marketing activity in other parts of the mix. Therefore, the more complete the regulation on marketing activities, the easier it will be to implement and the more effective it will be in reducing alcohol-related harm. Potential strategies leading to restrictions on promotion of alcohol products include warning messages, restricting surrogate marketing and total ban or comprehensive restrictions incorporating all forms of new and emerging media.

Restrictions on physical availability of alcohol include preservation of alcohol monopoly (if existing), licensing of alcohol sales points, regulation of alcohol outlets density, minimum legal age for alcohol purchase on and off premise, regulation for remote (online) selling.

A Return on investment of the three best buys implementation is $9 for every $1 invested [23]. Over 50 years, a 20% global increase in alcohol taxes alone could avert nine million premature deaths [22]. Revenues from excise tax, alcohol company taxes, and licensing fees could also help cover, or even meet, the costs of a comprehensive alcohol control programme, the

**Table 6.**

| SDGs | Alcohol adversely affecting SDGs |
|------|----------------------------------|
| SDG 1 | • A study in Sri Lanka found that over 10% of male respondents reported spending as much as or more than their regular income on alcohol |
| SDG 2 | • Especially in poorer communities, in families affected by alcohol use disorder, and in Low- and Middle-Income Countries (LMICs), alcohol tends to crowd out other more productive household spending, for example on education, health care and healthy food. |
| SDG 3 | • Alcohol is widely established as a structural driver of both the tuberculosis and HIV/AIDS epidemics.<br>• Alcohol is a major risk factor for NCDs, including mental ill-health.<br>• Globally, alcohol causes a net cardiovascular disease (CVD) burden of 593000 deaths.<br>• Worldwide, alcohol is responsible for 7.2% of all premature mortality. |
| SDG 4 | • Easy and wide availability of alcohol and other drugs, social norms permissive to substance use and detrimental to academic achievement, lack of positive contact with other adults. |
| SDG 5 | • The WHO Global Plan of Action on interpersonal violence identifies "ease of access to alcohol" as a risk factor for the occurrence of gender-based violence, including against children. |
| SDG 6 | • To get one litre of wine, 870 litres of water are needed.<br>• Per one litre of beer, 298 litres of water have to be used. |
| SDG 8 | • Cost of alcohol harm in European Union was €156 billion yearly.<br>• The economic burden of alcohol worldwide is substantial, accounting for up to 5.44% of Growth Domestic Product (GDP) in some countries.<br>• In the UK, as many as 89 000 people may be turning up to work hungover or under the influence of alcohol every day. The cost to the economy is up to £1.4 billion. |
| SDG 10 | • In the UK, health inequalities are estimated to cost £32–33 billion per year.<br>• Increasing alcohol taxes will avert 9 (20% increase) to 22 (50% increase) million premature deaths over a 50-year period. |
| SDG 11 | • Cali, Colombia: Closing alcohol outlets two hours earlier reduced homicides by 25%. |
| SDG 12 | • By some estimates, up to 92% of brewing ingredients are wasted. |
| SDG 13 | • In a lifecycle analysis of a Spanish beer, production and transport of raw materials used in beer production was found to contribute over one third of the total global environmental impact of the beer production lifecycle. |
| SDG 15 | • Often permissions for alcohol production are granted without adequate environmental impact studies. |
| SDG 16 | • USA: 1% increase in state-level excise beer tax resulted in a 0.3% reduction in child abuse rates and a 3% reduction in domestic abuse. |
| SDG 17 | • There is an inherent conflict of interest between Big Alcohol's goals (promoting consumption of products that harm health, economy, social fabric and the environment) on one hand and the SDGs on the other hand. |

Selected examples of interaction between consequences of alcohol use and sustainable development goals [7].

prevention and treatment of disorders caused by alcohol use, as well as contributing to the funding of other health and development priorities.

## Limitations

**Assessment of the reviews, not of the actual situation.** One limitation is that VNRs do not necessarily present the actual situation of implementation of various alcohol policy measures in the countries. We only assessed information provided in the VNRs. As can be seen from the volume of each review, it was up to each country to determine how much they chose to report. Moreover, the content of the VNRs could be affected by the fluctuating situation in countries. For example, during their VNR reporting, Estonia was in a period of high government attention to alcohol policy and new legislation was adopted. Similarly, Lithuania was in the process of adopting one of the most comprehensive alcohol policies in Europe. On the

other hand, Norway, also known for a well-developed alcohol policy system that has been in place for many years, was among the nine countries that have not scored on the initial key word search of alcohol and did not dedicate any space in their VNR to work on preventing and reducing alcohol harm.

**Subjectivity of the VNR assessment process.** Although the 260 questions form the base of an objective categorisation of any alcohol related mentions in the VNRs, there is a certain level of subjectivity in deciding how to interpret certain texts within the context of the VNRs. For example, some statistical information could be assigned only to the "indication" category but could as well be seen as reported impact of some policy measures. Similarly, interpretations of impacts beyond the SDG 3 were, in some cases, of subjective character as the reviews have not explicitly specified which other goals were positively or negatively affected by the actions taken.

## Implications

Although the potential of the alcohol policy best buys has not been reflected in countries' actions to achieve the SDGs as reported through the respective VNRs, alcohol is an obstacle to development. Therefore, not to consider the potential of the alcohol policy best buys in promoting sustainable development means a missed opportunity to achieve co-benefits across multiple SDGs in a cost-effective way. Countries are lagging behind the goals that they have committed to achieve. That is why every evidence-based and cost-effective action yielding results should be considered. Moreover, the cross-sectoral nature of the Agenda 2030 should be remembered when planning or adjusting the next implementation steps. Alcohol use is, for example, associated with violence (SDG 5 and 16), contributes to inequalities (SDG 5 and 10), undermines economic growth (SDG 8), disrupts sustainable consumption (SDG 12) and adversely impacts the environment (SDG 6 and 15). The findings of this study indicate that these effects are not considered in the design of measures to achieve the SDGs.

Furthermore, as our findings show, the analysed countries are often not precise in their definition of alcohol harm. This contributes to insufficient alcohol policy action and often to the choice of ineffective measures, as presented in the reviews. Awareness raising campaigns focused on individual behaviour change are neither in line with the WHO Global Alcohol Strategy that all selected countries adopted in 2010, nor do they reflect the magnitude of alcohol harm in these countries.

As time until 2030 is short and the coronavirus pandemic has further heightened the need for action, the challenge for all governments is to make the most of the potential afforded by the alcohol policy best buys to address alcohol related harm and in doing so improve outcomes in multiple SDG goals and targets.

## Supporting information

**S1 File.**
(XLSX)

## Author Contributions

**Conceptualization:** Kristina Sperkova, Eva Jané Llopis.

**Data curation:** Kristina Sperkova.

**Formal analysis:** Kristina Sperkova.

**Supervision:** Eva Jané Llopis.

**Writing – original draft:** Kristina Sperkova.

**Writing – review & editing:** Kristina Sperkova, Peter Anderson.

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
