## [Decision Letter · Decision Letter 0]

17 Feb 2022

PONE-D-21-32004Alcohol policy measures are an ignored catalyst for achievement of the Sustainable Development GoalsPLOS ONE

Dear Dr. Kristina,

Thank you for submitting your manuscript to PLOS ONE. After careful consideration, we feel that it has merit but does not fully meet PLOS ONE’s publication criteria as it currently stands. Therefore, we invite you to submit a revised version of the manuscript that addresses the points raised during the review process.

We look forward to receiving your revised manuscript.

Kind regards,

Prakash Kumar Sarangi

Academic Editor

PLOS ONE

Journal Requirements:

Additional Editor Comments:

major revision

Reviewers' comments:

Reviewer's Responses to Questions

**Comments to the Author**

1. Is the manuscript technically sound, and do the data support the conclusions?

Reviewer #1: Yes

Reviewer #2: Yes

2. Has the statistical analysis been performed appropriately and rigorously? 

Reviewer #1: No

Reviewer #2: I Don't Know

3. Have the authors made all data underlying the findings in their manuscript fully available?

Reviewer #1: Yes

Reviewer #2: Yes

4. Is the manuscript presented in an intelligible fashion and written in standard English?

Reviewer #1: Yes

Reviewer #2: Yes

5. Review Comments to the Author

Reviewer #1: 1. There are some papers/ reports/ online information on the same topic and, therefore it is not clear how this paper is distinguished from the previously published papers/ reports/ online information and what is the overall novelty of this paper. This must be clearly outlined in the 'Introduction'.

2. In discussion section, based on present study, elaborate (as far as possible) in detail regarding all potential strategies that can have promising viability in terms of implementation of alcohol policy measures for achievement of the Sustainable Development Goals.

Reviewer #2: Alcohol policy measures are an ignored catalyst for achievement of the Sustainable

Development Goals is good work done by the author. However, it needs little more discussions.

The interpretation of the data and the results could be better.

Results can be discussed in a better way.

Proper referencing is required in the methods section.

6. PLOS authors have the option to publish the peer review history of their article (what does this mean?). If published, this will include your full peer review and any attached files.

Reviewer #1: **Yes: **Dr. Akhilesh Kumar Singh, Mahatma Gandhi Central University, Bihar, India

Reviewer #2: No

---

## [Author Response · Author response to Decision Letter 0]

11 Mar 2022

Dear Prakash and dear reviewers,

thank you very much for your valuable comments. Please find our responses below.

Kind regards,

Kristina

Reviewer #1: 1. There are some papers/ reports/ online information on the same topic and, therefore it is not clear how this paper is distinguished from the previously published papers/ reports/ online information and what is the overall novelty of this paper. This must be clearly outlined in the 'Introduction'.

Response:

We have divided available related information into three categories and described the focus of the studies/reports/ belonging into these categories. (That has also led to adding two more sources to the Bibliography).

1. Problem description – alcohol harm is an impediment to achievement of sustainable development

2. Analysis of policy making in general and the Agenda 2030 in particular characterized by policy incoherence caused by emphasizing public health and at the same time furthering trade and partnership of governments with the private sector, including industries with vested interests.

3. Description of barriers that prevent making alcohol policy the priority in should be in the design of policies to achieve sustainable development

Then we have described the focus of this paper that is different from the literature available. To our knowledge, there is no similar paper that would offer a systematic overview of how governments report on their use of alcohol policy solutions in their work with Agenda 2030.

Reviewer #1: 2. In discussion section, based on present study, elaborate (as far as possible) in detail regarding all potential strategies that can have promising viability in terms of implementation of alcohol policy measures for achievement of the Sustainable Development Goals.

Response:

We have presented the three alcohol policy best buys and implementation actions/strategies they include as well as benefits of their implementation. We have also inserted an overview of selected examples showing adverse effect of alcohol use on achievement of SDGs. A complete overview of and a deeper elaboration on the impact of each alcohol policy measure on achievement of a particular SDG would be a separate paper. 

Reviewer #2: Alcohol policy measures are an ignored catalyst for achievement of the Sustainable

Development Goals is good work done by the author. However, it needs little more discussions.

The interpretation of the data and the results could be better.

Results can be discussed in a better way.

Proper referencing is required in the methods section.

Response: 

In the discussion and interpretation

- We expanded the reflection on implementation strategies of the three alcohol policy best buys. 

- We reflected on the nature of solutions countries discuss in their VNRs – population-wide vs individual responsibility

- We reasoned why the lack of consideration of alcohol policy best buys is a missed opportunity.

- We added a reason to the lack of consideration of alcohol policy solutions

- We added possible country response to these reasons

In the results we added a text about the importance of the 3.5.2 indicator, we elaborated on intermediate impact, impact of the SDG3 on other goals and we have added a reflection on the impact of “SDGs in focus” on reporting and of the report length on the level of elaboration.

Referencing in the methods section has been improved (references to HLPF, VNRs)

---

## [Editor Report · Decision Letter 1]

1 Apr 2022

Alcohol policy measures are an ignored catalyst for achievement of the Sustainable Development Goals

PONE-D-21-32004R1

Dear Dr. kristina,

We’re pleased to inform you that your manuscript has been judged scientifically suitable for publication and will be formally accepted for publication once it meets all outstanding technical requirements.

Kind regards,

Prakash Kumar Sarangi

Academic Editor

PLOS ONE

Additional Editor Comments (optional):

accepted
---

## [Editor Report · Acceptance letter]

8 Apr 2022

PONE-D-21-32004R1 

Alcohol policy measures are an ignored catalyst for achievement of the Sustainable Development Goals 

Dear Dr. Sperkova:

I'm pleased to inform you that your manuscript has been deemed suitable for publication in PLOS ONE. Congratulations! Your manuscript is now with our production department. 

Kind regards, 

on behalf of

Dr. Prakash Kumar Sarangi 

Academic Editor

PLOS ONE